# Mediation-Moderation Model: An Empirical Examination of Sustainable Women Entrepreneurial Performance towards Agricultural SMEs in Ivory Coast

Ingrid R. Epezagne Assamala [1], Wenyuan Li [1,*], Sheikh Farhan Ashraf [1,*], Nausheen Syed [2], He Di [1] and Mehrab Nazir [3]

1   School of Management, Department of Management Science & Engineering, Jiangsu University,
    Zhenjiang 212000, China; epezagneingrid@yahoo.fr (I.R.E.A.); hd@ujs.edu.cn (H.D.)
2   Department of Business Administration, Government College Women University, Faisalabad 38000, Pakistan;
    nausheen.dr@gmail.com
3   School of Economics and Management, Jiangsu University of Science & Technology, Zhenjiang 212000, China;
    mehrabnazir9@gmail.com
*   Correspondence: liwenyuan8@ujs.edu.cn (W.L.); fhsheikh08@gmail.com (S.F.A.);
    Tel.: +86-186-51404595 (S.F.A.)

**Abstract:** The consistent finding of knowledge management practices with women's entrepreneurial performance is one of the gaps intended to fill in this paper. Several previous research studies on knowledge management practices (KMPs) with sustainable women's entrepreneurial performance (SWEP) have been drawing the attention of many researchers, and this study includes the mediating role of opportunity recognition (OR) and moderating role of family interference (FI). Smart-PLS software was used to test the proposed hypotheses with gathered data of 450 women-entrepreneur respondents. The findings show a positive and significant impact of KMPs on women's entrepreneurship and partial mediation of opportunity recognition between sustainable women's entrepreneurial performance and KMPs. Moreover, family interference strengthens the relationship between opportunity recognition and women's entrepreneurial performance. The study results proposed that in Ivory Coast, entrepreneurial women face several challenges in running their businesses. Therefore, it is suggested that a combined effort of women entrepreneurs, family interference, society, market opportunities, and researchers can overcome their challenges. Discussion is based on the study findings, and suggestions have been made for researchers and practitioners.

**Keywords:** knowledge management practices; agriculture sector; sustainable women entrepreneurship; opportunity recognition; family interference; Ivory Coast

## 1. Introduction

Based on the facts and research, it is to acknowledge that SME entrepreneurs are the backbone of every country. However, most of the economies (emerging and emerged) do not much recognize women's entrepreneurial contribution to the country's growth. Women play a significant role and specific portion in economic development but are notably marginalized. Nowadays, the Ivory Coast government is focusing on the growth of women entrepreneurs, and many societies are dealing with women as second-class citizens [1]. Women entrepreneurs are not much successful compared to men entrepreneurs in terms of profitability and growth of a business [2]. However, women entrepreneurs struggle harder than men to avail all possible market and financial opportunities [3]. Several researchers confirmed that women could run the business, support the family, and give essential economic growth more efficiently than men [4]. This shows that women as entrepreneurs can play a significant role in recovering the loss of business society and nation.

Ivory Coast women face several challenges as compared to male entrepreneurs. There is 52% of women in Ivory Coast, but very less percentage is involved in business

due to culture. In Ivory Coast's society, women are being respected in routine domestic activities such as animal look after, food production, raising children, and cooking. Typically, women entrepreneurs compared weak compared to men entrepreneurs towards entrepreneurial activities [5]. Nevertheless, few women broke the barrier and started their business and present almost 46.2% of entrepreneurial activities to the economic growth [6]. In rural areas, women entrepreneurs are less in progress as compared to urban areas as well as men entrepreneurs [7]. There is a significant change in women's entrepreneurship performance, although they face huge challenges for their success. This study will help in highlighting the barriers faced by women entrepreneurs in doing their business, as well as possibly indicate ways to avert the problems.

The existing literature indicates that women-owned businesses performed slightly better than men while others do not find gender-based differentials in entrepreneurial performance. Indeed, controversy has arisen that women's businesses have a higher rate of failure or lower performance than men's businesses. Thus, it is the researcher's concern to study the key factors contributing to the significant results to the women entrepreneurs' performance. Towards the SME sector, most nations consider the significant role of women's entrepreneurial performance in economic growth. In 2012, Global Entrepreneurship Monitor (GEM) reported an estimation that 126 million women have started or running new businesses in around 67 countries around the world. In addition, about 98 million women are operating established businesses (www.gemconsortium.org accessed date: 15 December 2021).

Although in the last few years, KMPs have become quite an important line of research, which play an important role in organizational and economic growth. The existing literature explains the concept and importance of KMPs with entrepreneurial and firm performance. The researchers [8] defined the concept of KMPs in creating distinctive competencies in the organization. It is a managerial system that deals with the organizational and entrepreneurial inner structure for practical methodology and performance.

The knowledge-based economy creates a massive research culture for researchers and is seen as a key scope in economic growth [9]. Knowledge is an important key factor for entrepreneurs performing and accomplishing the task. Knowledge is a fundamental component to achieving competitive advantages and better performance [10]. KMPs assist in controlling the critical challenges towards competitive advantages in an organization. The culture of sharing knowledge shapes the organizational structure by exchanging knowledge, ability creation, innovation, and knowledge incorporation [11]. Consequently, all these resources determine knowledge management practices that ultimately change and accomplish business. Meanwhile, it is striving to find ways for companies to support their employees with knowledge resources to meet organizational challenges and improve performance in competitive markets [10,12].

The non-business women often cause anxiety in society, and women always struggle to define their roles and meaning in the world [13]. Therefore, female entrepreneurship is one of the most important means of establishing empowerment and improving the quality of women's life [14,15]. In entrepreneurial activity gender gap defines the difference between men and women performing activities [16]. A surprising number of studies focused on male entrepreneurs and few on gender differences of male-female entrepreneurship [17]. In preceding literature, researchers investigated the function of KMPs in overall entrepreneurial performance and the determined relationship between business and performance.

Agribusiness is the key to Ivory Coast's economic advancement, and more than 40% of agricultural African women entrepreneurs are engaged [18]. Sub-regional data can show that a significant proportion of women engaged in the agriculture sector. In 1980 the number of female entrepreneurs was 45%, and with time, it increased to 50% of the total population in African countries. However, women in Ivory Coast make up 49.6% of the total population, and due to gender bias, the proportion of women entrepreneurs is very less. Entrepreneurship is a key pillar of a nation's economic evolution and development as it helps reduce unemployment, improves living standards, and reduces poverty. Globally, social progress has made female entrepreneurial achievements visible [16].

However, female entrepreneurs have entered the labor market and started businesses to become independent [19], emphasizing the existence of female entrepreneurs [20]. Most studies focus primarily on European women entrepreneurs, with a more individualistic culture than Ivory Coast, where people share strong ties. Many of the research centers on female entrepreneurship [21], no specific studies examine the cultural and family interference in the context of the Ivory Coast. The study intends to deepen the fundamental problem of women entrepreneurship, why are males more entrepreneurs than females? Female entrepreneurs are mostly engaged in two different roles, work and family, which they also have to consider with their business [14,22].

Women's entrepreneurial performance is directly associated with knowledge management practices; both SWEP and knowledge management practices play a positive role in ensuring business growth [23], and the women's entrepreneurial performance support and retains the organizational market value. However, fundamentals of KMPs, such as knowledge sharing capacity (KSC) and innovation capacity (IC), are directly related to the success of an entrepreneur [24]. The exchange of skills, information, and experiences within an organization enhances the knowledge-sharing capacity of an entrepreneur. The use of depositories and repositories is to grab the internal and external information with knowledge sharing capacity [25]. The industrial revolution and innovation have become the key source of competitive advantages.

Despite many problems, innovation has become imperative for the firm and entrepreneurial performance. With innovative strategies, the entrepreneurs can realize and attain high profit and market share successfully. Innovation positively affects entrepreneurial success using several environmental and contextual factors in an organization. To capture the market share, it's not necessary to focus only on pricing strategy; the existence of opportunities, capabilities, and family interference is also very important for better performance [26]. The ability to adopt innovation is important when the market opportunities change rapidly and constantly. Previous studies [27] sought that innovative capacity influences women's entrepreneurial performance with an enterprising spirit and driving forces. [28] considered that entrepreneurial age and experience are important factors in exploring innovation capacity.

Opportunity recognition contributes to competitive advantage and superior performance [29,30]. SMEs women are heavily reliant on opportunities to avail themselves, survive and succeed. Previous research indicates that SMEs should proactively search and identify opportunities for better success [31]. Thus, opportunity recognition profoundly impacts the women's SME performance, particularly when the target is clear [32]. In the agricultural sector, women entrepreneurs need to identify the opportunities for sustainable performance [33].

Addressing business, obligations, and family interference is always a challenge for female entrepreneurs, leading to confusion between work and family. Businesses with family interference are categorized as business-to-family interference. Females often need to divide their time and energy into several roles, which burdens female entrepreneurs and reduces the time required to succeed in a company and family.

In Ivory Coast's agricultural sector, female needs to recognize and acquire awareness about possible opportunities for sustainable performance [34,35]. In addition, [10] most researches investigate the role of knowledge management practices with women entrepreneurial performance in developed countries but very less with the family interference parameter in developing countries. Additionally, family interference is negatively related to safety compliance participation and affects women's entrepreneurial performance [36]. It also identifies that family interference could be high risk in occupational injury for women entrepreneurs.

This research has some significant contributions to theory and practices. First, our study illustrates the concept of knowledge management practices with sustainable women entrepreneurial performance literature. Second, the mediating role of opportunity recognition leads to sustainable women entrepreneurial performance, and it has identified the

intermediary link between knowledge management practices with sustainable women entrepreneurial performance [37]. Third, family interference is exerted to the sustainable women's entrepreneurial performance as a moderator. Knowledge management practices acquire internal and external knowledge to motivate the entrepreneur to take the initiative and develop KMPs that improve entrepreneurial performance.

Thus, investigating a complementary perspective would fill a research gap, and this study covers the existing gap in the literature of knowledge management practices towards sustainable women's entrepreneurial performance. There have been no formal studies that examine the impact of combining the concepts of knowledge sharing capacity innovation capacity to achieve sustainable performance [37]. The relevance of opportunity recognition in the relationship between dynamic capability and long-term entrepreneurial performance has also been overlooked in prior studies. Section 2 covers the literature and hypotheses; meanwhile, Section 3 includes research framework research methodology; Section 4 presents results of the analysis. Finally, Section 5 describes the discussion and implication, followed by conclusions.

## 2. Literature Review and Hypotheses Formulation

Knowledge management practices are the scientific and systematic workforce planning for the operational and strategic needs of an organization. The relevance of this study lies with the fact that its findings will stimulate and strengthen the sustainability of women's entrepreneurial performance concerning knowledge management practices. This is a tough task for the new women generation to work and accustomed to existing SME culture and strategy. The entrepreneur needs to rethink strategies to keep pace with the changing trends of KMPs in the new era to acquire the best capacities and opportunities. It is important to implement a flexibility structure of knowledge sharing in an organization that will enhance the innovation capacity and women's entrepreneurial performance motivate [38].

Implementation of KMPs in an organization leads to entrepreneurial performance. A knowledge-sharing culture with positive behavior builds the learning capacities of an entrepreneur through experience, expertise, and innovation. KM practices significantly impact the organizational culture and entrepreneurial performance [39]. Entrepreneurs having a culture of sharing knowledge in their organization leads to the smooth functioning of knowledge flow, integration, and innovation capabilities. The KM practices stimulate the entrepreneurs to move beyond their comfort zone and rethink something new about the smooth functioning of the organizational and entrepreneurial performance.

Rigorous knowledge creation and dissemination impart the women entrepreneur for their inherent quality and make something new. Performance is encountered concerning quality. KM practices distinguish the organization from others and are a critical resource for gaining and sustaining successful business performance [40]. The model identified two major enablers of KMPs, namely, knowledge sharing capacity and innovation capacity. Knowledge sharing capacity comprises multiple dimensions such as technology, structure, and culture of the organization, whereas knowledge process capability comprises acquisition, conversion, application, and protection of the information [41]. The purification of knowledge, identification, creation, assimilation, and evaluation process is regarded as knowledge sharing capacity.

Knowledge management stands for a solution that streamlines the process of knowledge collection, distribution, and effective use [10,42]. The ability to share knowledge helps an entrepreneur make decisions on time [43]. Knowledge is living inside the individual's thoughts. Simultaneously, its miles elaborate, capture, disseminate, documentation, products, services, centers, and structures will become part of the business [44]. The resource-based theory (RBT) is useful for a theoretical framework to understand how competitive advantage is achieved within companies and how that advantage can be maintained over time [45]. It has become one of the critical theoretical perspectives with extensive acceptance in strategic management [46].

However, traditionally, market-based approaches focused on external factors, opportunities, threats, internal source strength, and weaknesses in entrepreneurial performance to achieve a competitive advantage. The resource-based theory (RBT) is an appropriate logical framework to understand how competitive advantage is achieved and sustained within the companies over time [45]. Knowledge is a strategic organizational resource that guides entrepreneurs to achieve sustainable performance. Hence, the entrepreneur should emphasize developing market knowledge for better performance [47].

The knowledge management practices introduce and acquire organizational understanding [48]. These practices perform together in understanding, sharing, and acquisition, contributing to innovation and enhancing entrepreneurial performance. The manner of understanding practices in a business is complicated, and the marketers are managing, respectively. Therefore these studies spotlight the principle additives to enhance the understanding of sustainable entrepreneurial performance. The study was conducted to test and identify the proposed model through appropriate research methods. There is no empirical study that describes the whole model with appropriate methodology. At the minimum, the research method used to formulate the collected data, research design through structural equation model. This study partially solved the research problems by evaluating the direct relationships of all studied variables in the proposed research model.

### 2.1. Relevance with Knowledge Sharing Capacity, Opportunity Recognitio, and Sustainable Women Entrepreneurial Performance

A firm's knowledge base perspective is established in the strategic management literature [49]. The resource-based theory can extend and identify opportunities for entrepreneurs to improve performance [50]. Organizations and entrepreneurs often rely on external sources to drive creativity and ameliorate performance [51–53]; as a result, if observed, shared knowledge does not work well, it affects the performance badly [54,55]. Opportunity recognition has been highlighted as a key contributor to entrepreneurial performance. SME entrepreneurs rely much on opportunities for survival and success [56]. Extant studies have provided abundant support to the positive relationship between opportunity recognition and SME performance.

This study explores the mediating role of opportunity recognition in the relationship between knowledge management practices and SME sustainable women entrepreneurial performance. Thus, before hypothesizing the mediating effect of OR, we build the direct relationship between opportunity recognition and SME performance, along with KMPs with SWEP. For instance, [29] found that a venture's performance is associated with its founders' ability to recognize an opportunity. The entrepreneur focused on opportunities for new ventures to grow and increase sales [31]. Thus, this research does;

**Hypothesis 1a (H1a):** Knowledge sharing capacity has a significant impact on opportunity recognition.

**Hypothesis 1b (H1b):** Knowledge sharing capacity significantly impacts sustainable women entrepreneurial performance.

**Hypothesis 1c (H1c):** Opportunity recognition mediates between knowledge sharing capacity and sustainable women entrepreneurial performance.

### 2.2. Innovation Capacity, Opportunity Recognition, and Sustainable Women Entrepreneurial Performance

Based on existing research, gender entrepreneurship is extremely extensive. There is widespread consensus that men start their businesses on a larger scale and have a greater trend in the male group. The RBT is most used for gender entrepreneurship performance. According to the business point of view, women are more likely to be compassionate and obedient [57]. Women entrepreneurs are more in innovating new and different things using

different tools to combine vision and skills to develop new ideas [58,59]. Each organization plans to start a new business with an innovative approach. The challenge is not only to reveal innovation capacity also to explore opportunities to support business. To innovate, an entrepreneur needs to identify new opportunities and coordinate resources to capture opportunities for better performance [60]. Thus, innovation capacity impacts the opportunity recognition in the SME sector and affects the performance positively. Opportunity recognition is highly related to performance, where huge product innovation capacities are largely involved.

Innovation capacity enables entrepreneurs to recognize opportunities for further achievements. We thereby argue that opportunity recognition mediates the relationship between knowledge management practices and women's entrepreneurial performance for the following reasons. First, opportunity recognition can facilitate innovation capacity. It also enables an entrepreneur to identify opportunities to take advantage, as the innovation capacity highlights the better options to exploit opportunities to perform better. Hence, building on the orientation-action-outcome framework, we test the mediations of opportunity recognition with innovation capacity and sustainable women's entrepreneurial performance. In this regard, we elaborated the following hypotheses:

**Hypothesis 2a (H2a):** Innovation capacity has a significant and positive influence on opportunity recognition.

**Hypothesis 2b (H2b):** Innovation capacity impacts positively on sustainable women entrepreneurial performance.

**Hypothesis 2c (H2c):** Opportunity recognition mediates positively between innovation capacity and sustainable women entrepreneurial performance.

*2.3. Opportunity Recognition and Sustainable Women Entrepreneurial Performance*

Entrepreneurs find possibilities from the market to perform [34,61]. Recognizing possibilities is a cognitive procedure based on the potential of human beings to apprehend styles and join the dots [62]. With transforming current information into innovation, the entrepreneur must explain the ideas primarily based on records. To give shape, such records are important to creating new commercial enterprise possibilities [63]. Moreover, few researchers declare that to perceive opportunities, the entrepreneurs seek procedures and discover different business tools for performance. Thus, entrepreneurial opportunities may be diagnosed and built simultaneously [64]. This possibility can arise at the start of a business; however, it can arise in the lifetime [10,65]. Entrepreneurs recognize opportunities to utilize as per the available resources [61,66]. Entrepreneurs create opportunities after interacting with others in bouncing thoughts again and forth. Creating possibilities for performance is a social procedure [20,66,67] and is based on the entrepreneur's potential to interact [68]. Opportunity recognition impacts women entrepreneurs positively for a firm's success [35]. As the opportunity recognition on time, always enhance the entrepreneurial performance.

**Hypothesis 3a (H3a):** Opportunity recognition impact positively on sustainable women entrepreneurial performance.

*2.4. Moderating Role of Family Interferences in the Relationship between Opportunity Recognition and Sustainable Women Entrepreneurial Performance*

Female entrepreneurs encounter family obstructions that reduce the trade execution [69]. Women entrepreneurs tend to have restricted control over macro-level, standardizing desires more than men, and should allot more family needs than work [70]. In developing countries, the majority of the women are engaged, and bond for home routine work, and families in contrast with their entrepreneurship behavior. Both male and

female entrepreneurs may interfere with their lives due to their routine duties. However, this trouble seems to be moderated in men's lives and more noticeably felt by women entrepreneurs [71]. The women are trained to prioritize their families, very less families acknowledge females for business [13,72]. The negative role of family interference affects women's business and technical skills, education, and performance. The family interference strengthens the relationship of opportunity recognition with sustainable women's entrepreneurial performance.

**Hypothesis (H4a):** Family interference impacts positively on sustainable women entrepreneurial performance.

**Hypothesis (H4b):** Family interference moderates the relationship between opportunity recognition and sustainable women entrepreneurial performance.

### 2.5. Conceptual Frame

Figure 1 shows the conceptual model for the studied variables. The study examined the relationship and effects of knowledge sharing and innovation capacities on sustainable women's entrepreneurial performance. Moreover, the study also examined the role of opportunity recognition and family interference as mediators and moderators between KMPs and SWEP.

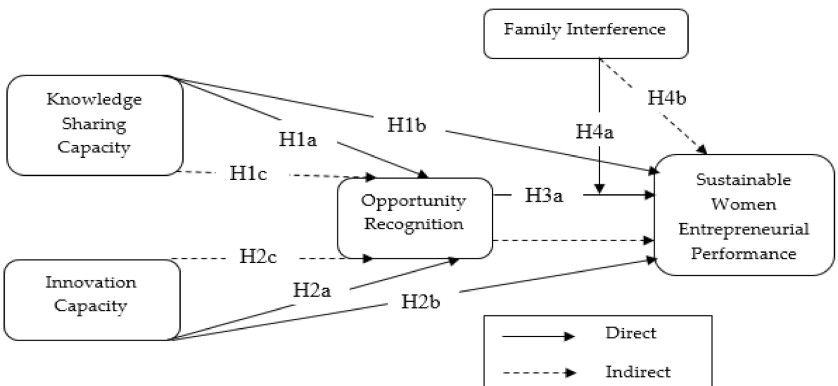

**Figure 1.** Conceptual framework.

### 3. Material and Methodology

To examine the proposed conceptual model quantitative approach was applied to numeric data collected from a large population. The analysis was applied to evaluate the reliability and validity of the gathered data. Prior research proposed that quantitative research is more appropriate to examine the relationship of latent constructs [73]. A deductive approach was applied, based on the existing resource-based theory (RBT) and a structured questionnaire using probability convenience sampling techniques [74]. The cross-sectional study was applied in September 2021 [75]. Some 650 questionnaires were distributed among SME women entrepreneurs; in December 2021, the process was completed with a return of 575, from which 125 were discarded due to missing information and 450 were used for further analysis. There was no designed information about women entrepreneurs in the chamber of commerce in Ivory Coast. It takes four months to complete the process due to the busy schedule of agricultural women entrepreneurs.

The author focuses on small-medium enterprises (SMEs) women entrepreneurs who mostly involve their family members and hire fewer employees in their businesses if needed. Ivory Coast is still very backward in SME women entrepreneurs, and very little research was conducted on the agricultural sector. Selecting an appropriate sample size for any research, the methodology is a crucial part; an inadequate sample size could lead to the study's failure or may not be truly representative of the population. To manage a large number

of data is not easy; it's expensive and time-consuming, which may affect the findings [76]. Although there are more than 70 languages in Ivory Coast, the author focused on cities where French is their mother language. The author contacted the respondents physically in different cities, such as Dabou, Alepe, Abengourou, Yamoussoukro, and Abidjan with a French questionnaire.

The questionnaire was drafted in French because of Ivory Coast's official and national language, so respondents can read and understand the questions easily [75].

Moreover, the researcher encouraged more females to participate in the survey for better results. This study is based on the women's sector as research aims to promote women SMEs. To evaluate the proposed hypotheses, the partial least square (PLS)–structural equation modeling (SEM) technique was applied using Smart-PLS v3. Smart-PLS is authentic software to test mediation–moderation models, and it also works for normal distributions simultaneously and multivariate [77].

### 3.1. Demographics

Table 1 shows the demographic for targeted respondents, including age, qualification, business sector, and duration. Most women entrepreneurs fall between the ages of 46 to 50 and above with 32.45 and 30.89 percent, which is very high. The young women entrepreneurs are very less in number with 9.11%. Women entrepreneurs were very few with professional and university level qualifications than the middle and high class. During the data collection process, women with less education are more successful than young and educated in the agricultural sector because of the leadership and consistency. The Ivory Coast agricultural sector involves several businesses, but the researcher focuses on five sub-units such as crop production, animal production, food services, forestry, agri-support services. Therefore, we considered all these units for data collection, and the percentages are presented.

**Table 1.** Sample statistic frequency distribution.

| Particulars | Description | Frequency | Percentage |
|---|---|---|---|
| Gender | Female | 450 | 100% |
| Age (in years) | 18–25 | 41 | 9.11% |
| | 26–35 | 59 | 13.11% |
| | 36–45 | 65 | 14.44% |
| | 46–50 | 146 | 32.45% |
| | 51 Above | 139 | 30.89% |
| Educational Qualification | Middle school | 218 | 48.44% |
| | High school | 106 | 23.56% |
| | Graduation level | 52 | 11.56% |
| | University level | 63 | 14.00% |
| | Professional education | 11 | 2.67% |
| Business Sector | Crop production | 150 | 33.33% |
| | Animal production | 77 | 17.11% |
| | Food services | 140 | 31.11% |
| | Forestry | 30 | 6.67% |
| | Agri-support services | 53 | 11.78% |
| Business Tenure | 1–5 years | 56 | 12.45% |
| | 6–10 years | 121 | 26.89% |
| | 11–15 years | 87 | 19.33% |
| | 16–20 years | 84 | 18.67% |
| | 21–25 years | 73 | 16.22% |
| | 25 years above | 29 | 6.44% |

*3.2. Measures*

This research was based on a structured questionnaire and designed to ensure the realistic and practical implications of the model. The questionnaire was adopted and adapted from existing studies for sustainable women entrepreneurial performance, knowledge-sharing capacity, innovation capacity, opportunity recognition, and family interference. To measure the composition, a 5-point Likert scale from 1 to 5 (1—Strong disagree, 5—strongly agree) was used to estimate the outcomes [78].

Appendix A shows the adopted and adapted questionnaire for all studied constructs. To measure knowledge-sharing capacity, five items were adapted from the study by Hsu [79] and validated by [80]. The researchers adopted five measuring scales developed by Hurley [81]. The six measurement items of opportunity recognition were measured from the [82]. The 11 items of entrepreneurial performance were adopted from the study of [83]. Family interference was measured using two-dimensional exploration and utilization in each of the three items, six items from the scale developed by [84]. Previous researchers used this scale.

*3.3. Measurement Model*

To test the convergent validity, Statistical Packages for Social Sciences (SPSS) and Smart-PLS software are used to minimize the flaws and fluency in data. SEM is applied to evaluate the empirical and causal model [85]. Smart-PLS calculate the path model for formative and reflective mode, multivariate analysis, moderating, and mediating model [86]. Table 2 describes the convergent validity for all studied variables and extracts the factor loading with a rule of thumb of at least 0.7. Convergent validity table also describes Cronbach's alpha value which should be $\geq$0.7, rho_A 0.7, the average value extracted (AVE) should be $\geq$0.5 [87], composite reliability should be $\geq$0.7 [88], and confirmatory factor analysis (CFA) was above thrush hold value and acceptable [89].

**Table 2.** Measurement model.

| Variables and Constructs | Loadings | CA | rho-A | CR | AVE |
|---|---|---|---|---|---|
| **Knowledge-Sharing Capacity** | | 0.946 | 0.949 | 0.947 | 0.782 |
| KSC1 | 0.929 | | | | |
| KSC2 | 0.916 | | | | |
| KSC3 | 0.880 | | | | |
| KSC4 | 0.839 | | | | |
| KSC5 | 0.907 | | | | |
| **Innovation Capacity** | | 0.928 | 0.929 | 0.926 | 0.716 |
| IC1 | 0.892 | | | | |
| IC2 | 0.846 | | | | |
| IC3 | 0.870 | | | | |
| IC4 | 0.881 | | | | |
| IC5 | 0.804 | | | | |
| **Opportunity Recognition** | | 0.945 | 0.947 | 0.945 | 0.741 |
| OR1 | 0.845 | | | | |
| OR2 | 0.791 | | | | |
| OR3 | 0.916 | | | | |
| OR4 | 0.959 | | | | |
| OR5 | 0.852 | | | | |
| OR6 | 0.895 | | | | |

**Table 2.** *Cont.*

| Variables and Constructs | Loadings | CA | rho-A | CR | AVE |
|---|---|---|---|---|---|
| **Sustainable Women Entrepreneurship Performance** | | **0.940** | **0.943** | **0.938** | **0.583** |
| **SEP1** | 0.720 | | | | |
| **SEP2** | 0.754 | | | | |
| **SEP3** | 0.830 | | | | |
| **SEP4** | 0.836 | | | | |
| **SEP5** | 0.756 | | | | |
| **SEP6** | 0.899 | | | | |
| **SEP7** | 0.794 | | | | |
| **SEP8** | 0.796 | | | | |
| **SEP9** | 0.783 | | | | |
| **SEP10** | 0.760 | | | | |
| **EP11** | 0.779 | | | | |
| **SEP1** | | | | | |
| **Family Interference** | | **0.937** | **0.938** | **0.936** | **0.709** |
| **FI1** | 0.835 | | | | |
| **FI 2** | 0.873 | | | | |
| **FI 3** | 0.813 | | | | |
| **FI 4** | 0.912 | | | | |
| **FI 5** | 0.857 | | | | |
| **FI 6** | 0.920 | | | | |

*3.4. Common Method Bias and Multicollinearity Test*

To avoid multicollinearity and determine variance inflation factor (VIF) and common method bias (CMB), the Harman test was applied. No issue is found on CMB if merged factors are less than 50% of the variance [90]. Table 3 shows that no value exceeds 10, which shows that there is no multicollinearity issue.

**Table 3.** Collinearity statistics (VIF).

| | |
|---|---|
| FI1 | 3.901 |
| FI 2 | 3.941 |
| FI 3 | 2.550 |
| FI 4 | 2.394 |
| FI 5 | 4.248 |
| FI 6 | 3.128 |
| SWEP1 | 2.072 |
| SWEP 10 | 2.571 |
| SWEP 11 | 2.621 |
| SWEP 2 | 2.596 |
| SWEP 3 | 2.985 |
| SWEP 4 | 2.586 |
| SWEP 5 | 2.010 |
| SWEP 6 | 3.218 |
| SWEP 7 | 3.290 |
| SWEP 8 | 2.755 |
| SWEP 9 | 2.074 |

**Table 3.** *Cont.*

|      |       |
| ---- | ----- |
| IC1  | 2.151 |
| IC2  | 2.458 |
| IC3  | 4.385 |
| IC4  | 3.958 |
| IC5  | 3.977 |
| KSC1 | 4.686 |
| KSC2 | 4.569 |
| KSC3 | 3.873 |
| KSC4 | 2.634 |
| KSC5 | 3.895 |
| OR1  | 2.641 |
| OR2  | 2.715 |
| OR3  | 3.124 |
| OR4  | 4.293 |
| OR5  | 4.572 |
| OR6  | 4.434 |

### 3.5. Discriminant Validity

Table 4 shows the Fornell–Larcker criterion and includes cross-loading values. Therefore, the table shows that there is no discriminant validity issue found.

**Table 4.** Fornell-Larcker criterion.

|          | FI    | IC    | KSC   | OR    | SEP   |
| -------- | ----- | ----- | ----- | ----- | ----- |
| **FI**   | 0.842 |       |       |       |       |
| **IC**   | 0.238 | 0.846 |       |       |       |
| **KSC**  | 0.272 | 0.412 | 0.884 |       |       |
| **OR**   | 0.400 | 0.323 | 0.387 | 0.861 |       |
| **SWEP** | 0.442 | 0.419 | 0.385 | 0.390 | 0.764 |

As per the rule of thumb for heterotrait–monotrait ratio (HTMT), the values should be 1 among all factors [91]. Table 5 shows the HTMT analysis, which explored the discriminant validity values [77], and the values were much closer. The below table shows that all values of HTMT are within the threshold values. Therefore, no discriminant validity issue in HTMT.

**Table 5.** Heterotrait–Monotrait (HTMT) ratios.

|          | FI    | IC    | KSC   | OR    | SWEP |
| -------- | ----- | ----- | ----- | ----- | ---- |
| **FI**   |       |       |       |       |      |
| **IC**   | 0.237 |       |       |       |      |
| **KSC**  | 0.270 | 0.409 |       |       |      |
| **OR**   | 0.400 | 0.322 | 0.387 |       |      |
| **SWEP** | 0.440 | 0.418 | 0.385 | 0.383 |      |

### 3.6. Structural Model for Sustainable Women Entrepreneurial Performance

As shown in Figure 2, Smart-PLS is used to apply the bootstrapping to extract the structural path model at 500 sub-samples. To analyze the fitness of the model, standardized root means square residual (SRMR) was applied, and the value for a good model should be <0.08.

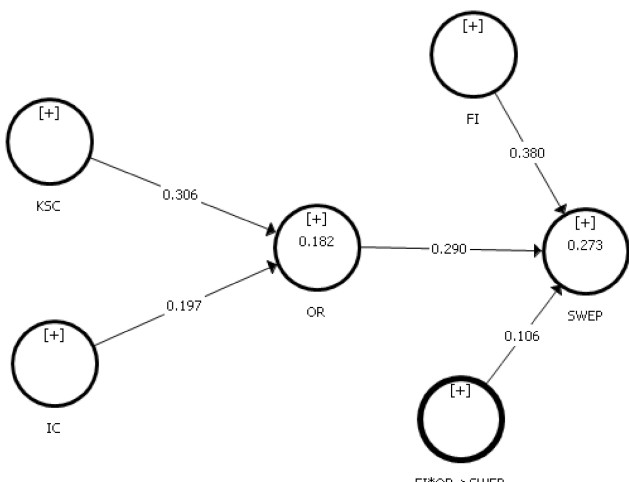

**Figure 2.** Path Model for SWE performance.

Table 6 shows the direct impacts such as; H1a KSC imapcts positively on OR, and supported by ($\beta$ = 0.306; t = 6.203; $p$ < 0.000). H1b demonstrate the positive effect of KSC on SWEP, and supported with ($\beta$ = 0.089; t = 3.391; $p$ < 0.001). H2a showed a direct impact of IC on OR, and supported with ($\beta$ = 0.197; t = 4.231; $p$ < 0.000). H2b outlined a significant, impact of IC on SWEP with ($\beta$ = 0.057; t = 2.886; $p$ < 0.004). H3a explained that OR affects on SWEP positively, and supported with the ($\beta$ = 0.290; t = 5.001; $p$ < 0.000). At the same time, H4a shows positve impact of FI on SWEP; therefore, H4a is supported with the values of ($\beta$ = 0.380; t = 6.138; $p$ < 0.000).

**Table 6.** Path Coefficients for direct relations.

| Hyp. | Relationships | $\beta$ | Mean | S.D | t-Value | *p*-Value | Decision |
|---|---|---|---|---|---|---|---|
| **H1a** | **KSC $\rightarrow$ OR** | 0.306 | 0.304 | 0.049 | 6.203 | **0.000** | Accepted |
| **H1b** | **KSC $\rightarrow$ SWEP** | 0.089 | 0.090 | 0.026 | 3.391 | **0.001** | Accepted |
| **H2a** | **IC $\rightarrow$ OR** | 0.197 | 0.201 | 0.047 | 4.231 | **0.000** | Accepted |
| **H2b** | **IC $\rightarrow$ SWEP** | 0.057 | 0.060 | 0.020 | 2.886 | **0.004** | Accepted |
| **H3a** | **OR $\rightarrow$ SWEP** | 0.290 | 0.294 | 0.058 | 5.001 | **0.000** | Accepted |
| **H4a** | **FI $\rightarrow$ SWEP** | 0.380 | 0.377 | 0.062 | 6.138 | **0.000** | Accepted |

*3.7. F-Square*

Table 7 describes the *f*-square values wherein a variable, and structural model may be affected by the number of variables. The change of R-square is F-square when the model removes an exogenous variable. It is used to measure the closeness of the predictors. If the value for *f*-square falls in $\geq$0.02 it's smaller, medium with $\geq$0.15, and having a larger value with $\geq$0.35 [92].

**Table 7.** Value of the F-square.

| | FI | FI*OR and SWEP | IC | KSC | OR | SWEP |
|---|---|---|---|---|---|---|
| **FI** | | | | | | 0.158 |
| **FI*OR and SWEP** | | | | | | 0.032 |
| **IC** | | | | | 0.040 | |
| **KSC** | | | | | 0.095 | |
| **OR** | | | | | | 0.092 |
| **SWEP** | | | | | | |

### 3.8. Cross-Validated Redundancy

Table 8 describes the Q-square values and explains the predictive relevance of the endogenous constructs. It is applied to measure the predictive relevance of the proposed model, and the thresh-hold value for $Q^2$ should be great than 0 and will describe the predictive relevancy of data. In Smart-PLS, blindfolding is applied for extracting the value of Q2.

**Table 8.** Construct cross-validated redundancy.

| Constructs | $Q^2$ (= 1 − SSE/SSO) |
| :---: | :---: |
| OR | 0.120 |
| SWEP | 0.137 |

### 3.9. Indirect Effects of Sustainable Women Entrepreneurial Model

Table 9 describes the path coefficient for indirect relation of knowledge sharing capacity and innovation capacity on sustainable women entrepreneurial performance through the mediation of opportunity recognition. Furthermore, moderation is also applied between opportunity recognition and sustainable women's entrepreneurial performance through family interference.

**Table 9.** Path coefficient for indirect relationship.

| Hyp. | Relationships | β | Mean | SD | t-Value | *p*-Value | Decision |
| :---: | :---: | :---: | :---: | :---: | :---: | :---: | :---: |
| H1c | KSC → OR → SWEP | 0.089 | 0.090 | 0.026 | 3.391 | **0.001** | proved |
| H2c | IC → OR → SWEP | 0.057 | 0.060 | 0.020 | 2.886 | **0.004** | proved |
| H4b | FI*OR → SWEP | 0.106 | 0.110 | 0.033 | 3.245 | **0.001** | proved |

Furthermore, H1c showed a partial mediating effect of OR between KSC and SWEP; and supported (β = 0.089; t = 3.391; $p < 0.001$). H2c also descriebe partial mediation of OR between IC and SWEP; and supported (β = 0.057; t = 2.886; $p < 0.004$). Moreover, the study also consider the moderating effect of FI on the relationship between OR and SWEP; with (β = 0.106; t = 2.245; $p < 0.001$).

## 4. Discussion

To explore the relation and impact of knowledge management practices (knowledge sharing capacity and innovation capacity) on sustainable women entrepreneurial performance, with the mediating role of opportunity recognition and moderating role of family interference between innovation capacity, knowledge-sharing capacity, and sustainable women entrepreneurial performance. Several analyses have been applied to proposed hypotheses and to perform path coefficient, supported empirically with significant findings, a *p*-value should be <0.05 and t-value > 2.

The proposed hypotheses and study findings explore the results for H1a is KSC has a significant and positive impact on opportunity recognition, the result is supported by [93]. According to existing literature and study findings, knowledge sharing capacity can bring valuable results by sharing information in the organization and entrepreneur for sustainable performance [94]. Sharing knowledge enhances the performance tendency within and outside the organization [95]. Knowledge-sharing capacity will demonstrate the entrepreneurs' opportunity recognition behavior and better opportunities which ultimately increase the entrepreneurial performance [96]. Timely opportunity recognition with capacities always enhances performance as H1b proposed that KSC impacts SWEP positively and is supported by the explanations in literature, also consistent with [97,98]. Knowledge sharing, competencies, and resources of an entrepreneur enhance performance. The H1c offers that opportunity recognition mediates the effect between sustainable women entrepreneurial performance and knowledge sharing capacity [98], which is consistent

with the study of [79]. The explanation for the mediating role of opportunity recognition increases the performance of a women entrepreneur [99].

H2a proposes that the impact of innovation capacity on opportunity recognition and findings are commented on by [81] and other studies by [100]. H2b findings with the proposed hypothesis that innovation capacity influenced sustainable women entrepreneurial performance, consistent with the prior studies of [101]. The findings of H2c confirmed the mediating role of opportunity recognition between innovation capacity and sustainable women entrepreneurial performance [102]. The study results contribute to the innovation capacity with opportunity recognition for sustainable women entrepreneurial performance [103]. Innovation capacity always supports the entrepreneurs to compete in the market and to avail competitive advantages. Innovation capacity with opportunity recognition boosts the decision-making power of an entrepreneur, which leads to better performance [31].

Furthermore, the results for H3a that opportunity recognition impact on women entrepreneurial performance and study findings are consisted by [104]. The study findings of H4a confirm that family interference impacts women's entrepreneurial performance positively and is consistent with [105]. The study findings for H4b reveal the moderating effects of family interference between opportunity recognition and entrepreneurial performance [106]. Moreover, the FI strengthens the relationship of opportunity recognition and sustainable women's entrepreneurial performance as a moderator. Furthermore, the study findings and existing literature also show the significant role of opportunity recognition as a moderator between KMPs and SWEP. It is suggested that OR is always intended to increase entrepreneurial performance.

## 5. Implications

### 5.1. Theoretical Implications

Current research results show an important relationship between innovation and knowledge capacity, awareness of opportunities, family intervention, and competitive advantages for sustainable women's entrepreneurial performance in the agricultural sector. Therefore, researchers need to build important relationships between these factors and identify why these factors are important to study. This study highlights the importance of the women's sector, which was ignored in most previous studies. We contribute to business research by exploring topics related to this group from various economic, social, and agricultural perspectives. This study covers the most important aspects of female entrepreneurs, lacking in research. This research allows other researchers to explore more about sustainable women's entrepreneurial performance. As a result of our research, female entrepreneurs have the opportunity to identify the strength and weaknesses of entrepreneurship, as well as existing opportunities and risks. This survey shows the female entrepreneurs' views on the internal and external factors that affect the organization. The study has a combination of theory and knowledge management practices which was not applied before in the agricultural sector in the Ivory Coast. This study will also help future researchers in similar fields, leading to better results.

### 5.2. Practical Implications

This research has implications for practitioners in the agricultural sectors, as well as academicians in the fields of female entrepreneurship and sustainable business accomplishment. First, this research participates in the literature on sustainable business performance, knowledge management skills, family intervention, and awareness of opportunities. To better understand the government and non-governmental agricultural sector of Ivory Coast, it is advisable to infer from this survey that it will help reduce the charts of failed businesses. The study will give much practical support to the poorly performing female entrepreneurs in the agricultural sector. This study will provide an effective way for female entrepreneurs in the agricultural sector to transfer knowledge within their organizations to build a sustainable environment for reaching business objectives against their competitors. Finally,

this research, in addition, contributes to the literature on knowledge management abilities to a wider perspective on the achievements of sustainable entrepreneurs by women.

## 6. Conclusions

This study contributes to the current literature by examining the importance of knowledge management practices, opportunity awareness, and family interference in improving the performance of female entrepreneurs in agribusiness. Therefore, female entrepreneurs in the agricultural sector need to focus on these factors to improve their performance. Goals are easy to achieve if female entrepreneurs are well-planned, motivated, and independent. The results showed that role of female entrepreneurs and knowledge management practices was recognized in opportunity detection skills to achieve adequate results. Awareness of knowledge and opportunities with positive family interference plays a key role in accomplishing tasks for female entrepreneurs.

This study showed that knowledge management practices regulate the performance of female entrepreneurs in the agriculture sector using significant beta factors, *t*-tests, and *p*-values. In addition, the results showed that awareness of opportunities plays an important role in women's sustainable business performance, and family interferences ease the relationship between awareness of opportunities and women's performance. These reasons show how knowledge management practices can help female entrepreneurs to perform well. This can positively impact the Ivory Coast's unemployment and economic growth of a country. This study informs the role of entrepreneurial performance and assessment of family interference.

These kinds of research are important and helpful in facing challenges due to family interference and family businesses. It increases the emotional burden on the owner and indirectly impacts performance by reducing happiness by allocating time, energy, and material resources to the company. Based on these considerations, our contributions are primarily aimed at scientists and practitioners who want to improve their study of female entrepreneurial performance.

Although female entrepreneurs have to face several challenges to perform better, such as improper guidance, no proper education, and technical skills to build infrastructure for business, most SME sector business people, especially in the agricultural sector, do not survive more than five years due to a lack of information, experience, and skills. It was surprising that most SMEs do not survive more than a few years. Here, knowledge management practices play an important role in learning, growing, and developing the agribusiness. Previous research explores the positive role of KMPs in the better performance of women entrepreneurs. Nowadays, the government of Ivory Coast has taken the initiative to support agriculture women entrepreneurs with field knowledge, advanced medication for crops, personal and economic growth. The proper guidance for women entrepreneurs in the agricultural sector will enhance their performance and give several kinds of benefits regarding business growth.

The government should open one window for female entrepreneurs, where they can acquire better suggestions for their businesses. It will also assist the star-ups, not to become lost in the business or less performance. There must be a trained staff with technical and business-oriented knowledge to conduct seminars and activities up to the level of women entrepreneurs for their better performance and growth. The agriculture sector always supports the country's economy, so the government should develop more pro institutes for women entrepreneurs brought up.

### *Limitations and Future Research Directions*

It must be acknowledged that this study has some limitations. The data was collected from one or the same source. The cross-sectional nature of the data is also limited, and longitudinal data is recommended to researchers for future research. As a direction for future research, this model will be useful in other research fields. More accurate and better conclusions for researchers can be seen as control variables for demographics, culture,

government policy, and regulation. As an additional limitation of the study, the sample population was gender-specific and consisted of 100% females. This transaction was based on a company in the female category. Therefore, gender composition should be considered for future study iterations to the current study. Finally, the proposed research model was tested by female entrepreneurs in Ivory Coast in the agricultural sector.

However, the survey may consider more different industries for future recommendations. Therefore, future researchers may carry out similar research patterns in different time frames. Therefore, a person's knowledge and learning ability can change over time. This is the main reason for suggesting future researchers carry out a longitudinal study of the ecosystem depicted in this study. Declaration of data availability, the author, provides raw data supporting the conclusions of this article without undue reservation.

**Author Contributions:** Conceptualization, S.F.A. and I.R.E.A.; methodology, I.R.E.A. and W.L.; software, N.S.; validation, M.N., H.D. and W.L.; formal analysis, I.R.E.A.; investigation, I.R.E.A. and H.D.; resources, W.L.; data curation, N.S. and M.N.; writing—original draft preparation, S.F.A.; writing—review and editing, S.F.A. and I.R.E.A.; visualization, M.N. and N.S.; supervision, W.L.; project administration, H.D. and W.L.; funding acquisition, I.R.E.A. All authors have read and agreed to the published version of the manuscript.

**Funding:** This research was funded by Epezagne Ngouandi and Epezagne Francis.

**Institutional Review Board Statement:** Not applicable.

**Informed Consent Statement:** Not applicable.

**Data Availability Statement:** Data will be provided on request.

**Acknowledgments:** We are grateful to Epezagne Ngouandi and Epezagne Francine for their immense support of this paper.

**Conflicts of Interest:** There is no conflict of interest while conducting research.

## Appendix A

**Table A1.** Questionnaire, participants were asked: "Please select a number from the scale below that best describes your response".

| Sr # | Questions | Responses | | | | |
|------|-----------|---|---|---|---|---|
| | **Entrepreneurial Performance** | 1 | 2 | 3 | 4 | 5 |
| 1 | Understands work responsibilities, scope of job tasks, and routines to be performed. | | | | | |
| 2 | Completes work thoroughly, accurately, and according to specifications. | | | | | |
| 3 | Avoids law or rules infractions, excessive absenteeism, or other behaviors that may have a negative impact on the organization or employees. | | | | | |
| 4 | Clearly and appropriately communicates information in writing. | | | | | |
| 5 | Clearly and appropriately communicates information orally. | | | | | |
| 6 | Contributes to the top management team by supporting other team members, resolving conflict between members, and contributing to general team functioning. | | | | | |
| 7 | Supports peers and performs cooperative, considerate, and helpful acts that assist coworkers' performance. | | | | | |
| 8 | Forms goals, allocates resources to meet them, and monitors progress toward them. | | | | | |
| 9 | Influences the performance of others in achieving the goals of the organization. Includes communicating goals to others, modeling appropriate behaviors, coaching others to help them attain goals, and providing reinforcement upon the attainment of goals. | | | | | |
| 10 | Overcomes natural resistance to organizational change; strives to behave in ways that are consistent with the change goals and corporate strategy. | | | | | |
| 11 | Effectively manages the transition period while organizational changes are being implemented. This involves dealing with the rate at which change is introduced and the processes used to introduce change. | | | | | |

**Table A1.** *Cont.*

| Sr # | Questions | Responses | | | | |
|---|---|---|---|---|---|---|
| | **Knowledge-Sharing Capacity** | **1** | **2** | **3** | **4** | **5** |
| 1 | I frequently participate in knowledge-sharing activities in the organization. | | | | | |
| 2 | I usually spend a lot of time on knowledge-sharing activities in the organization. | | | | | |
| 3 | When participating in the organization meetings I usually actively share my knowledge with others. | | | | | |
| 4 | When discussing complicated issues I am usually involved in the subsequent interaction. | | | | | |
| 5 | I usually involve myself in discussions of various topics rather than specific topics. | | | | | |
| | **Innovation Capacity** | **1** | **2** | **3** | **4** | **5** |
| 1 | Risk taking is encouraged in our firm. | | | | | |
| 2 | Creativity is encouraged in our firm. | | | | | |
| 3 | Management actively seeks innovative ideas. | | | | | |
| 4 | Management is tolerant of mistakes when taking risks. | | | | | |
| 5 | The firm is often first to market with new products and services. | | | | | |
| | **Family Interference** | **1** | **2** | **3** | **4** | **5** |
| 1 | I would put in a longer workday if I had fewer family demands. | | | | | |
| 2 | My family demands interrupt my workday. | | | | | |
| 3 | Family demands make it difficult for me to take additional entrepreneurial responsibilities. | | | | | |
| 4 | I spend time at work making arrangement for family members. | | | | | |
| 5 | Family demands make it difficult for me to have the work schedule I want | | | | | |
| 6 | When I am at work, I am distracted by family demands. | | | | | |
| | **Opportunity Recognition** | **1** | **2** | **3** | **4** | **5** |
| 1 | You carry out market research to identify new product/service ideas or new markets. | | | | | |
| 2 | You look for and maintain a good relationship with business colleagues to watch out for new business opportunities. | | | | | |
| 3 | Meetings with customers are important. | | | | | |
| 4 | You know that rapid changes in technology can affect your business. | | | | | |
| 5 | Your organizational structure is flexible to adapt to changes. | | | | | |
| 6 | You have enough experience to cope with the unexpected changes in the industry. | | | | | |

1—Strongly disagree, 2—Disagree, 3—Neutral, 4—Agree, 5—Strongly agree.

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
