# Peer review of "Mediation-Moderation Model: An Empirical Examination of Sustainable Women Entrepreneurial Performance towards Agricultural SMEs in Ivory Coast"

_sustainability, doi:10.3390/su14106368_

Round 1

Reviewer 1 Report

A Proof Reading is mandatory. A lot of sentences are incorrectly stated. For example: “The results partially mediate opportunities between women's sustainable entrepreneurial performance and knowledge management practices.” Are the results mediating????? Or : “It is effective for staying for a long time.” What? Who?

Introduction: a lot of rework must be conducted. This argument is flawless: “Most studies of female entrepreneurs focus primarily on Europe, where culture is individualistic, as opposed to Ivory Coast, a country where people share strong ties [18]. While many other past types of research have centered on the performance of female entrepreneurship [19], no specific studies are examining them due to their unique cultural and family intervention in the context of the Ivory Coast.” First, I’ve searched your citation [18] - Ventura, A.M., et al., Women’s Empowerment, Research, and Management: Their Contribution to Social Sustainability. 2021. 13(22): p. 12754., and I don’t see where they reveal that European culture is individualistic. Second, in your model you are not testing the “unique cultural and family intervention”, far from that.

By the way, most of the references are incomplete, so I can’t see whether you are basing you theory on good quality articles

The literature review is lacking for some support, I’m particularly not satisfied with the argumentation for H4 and H5. Inclusively, some sentences are tautological: “Both male and female entrepreneurs may have their lives interfered with by their work duties.”

Section 3.2. title is Measures not Measure. Note that you are not complying with the journal recommendation. You are saying that “Data Availability Statement: The datasets used in this study are available in the text and cited in the Reference section.” But the survey data is not in the sources indicated in the text.

This sentence makes no sense: “[89, 90] Via reliability and validity analysis, the model suitability was assessed.”

Results: must include: boostraping subsamples; Q2, VIF values. Table 2 must clearly include square root of AVE and the correlations.

Figure 2 and 3 are just a copy past from PLS showing a lack of concern with the reader. They are totally unformatted and the original colours are not welcome. Please rework the figure and include a legend with an explanation of the abbreviations

The discussion is not a discussion at all. This is another weak point. The authors must link their results with previous research, specifying where they align, contradict or advance existing knowledge. In case of advancing knowledge, the authors must recognize this also in the introduction and abstract. Note that this topic is mandatory. This is notorious in this part of the text were there is no value added, just a repetition of the hypotheses: “In addition, H3b: Absorptive Capacity positively influences the performance of female sustainable entrepreneurs, and the results are consistent with the current study H4a: Awareness of opportunities has a positive impact on the performance of female sustainable entrepreneurs. H4b: Opportunity awareness conveys the relationship between the ability to share knowledge and the performance of female sustainable entrepreneurs. H4c: Awareness of opportunity conveys the relationship between the ability to innovate and the performance of sustainable entrepreneurs by women. H4d: Awareness of opportunity conveys the relationship between female absorbency and sustainable entrepreneurial performance.

Author Response

I would like to say thanks to the reviewer for his precious time and valuable suggestions which really enhance the quality and credibility of this manuscript. The author tried to incorporate all the comments and on the other hand, the author revised the whole manuscript (Introduction, Literature, Hypotheses, Methodology, and Conclusion) parts. 

Reviewer 2 Report

The article is not well written. Many sentences cannot be understood or are unclear. This makes it impossible to assess the real value of the paper. The English is a barrier to the content.

As far as the content is concerned, the paper also has many problems.

The state of the art is not clearly given. It is not clear why such a study is needed. Mixing literature and hypothesis formulation makes this more difficult. The different aspects are separated, we do not get the full picture.

The items are not given in detail. As such, the reader cannot judge the outcome at all.

Sampling and recruiting remains unclear. How did you make sure that only women in agriculture participated? What is the size of the enterprises? Did they have employees or do they simply sell their own produce?

Maybe they are extremely heterogeneous. Then it might be highly questionable to mix them. Maybe they work in different agricultural domains. Some might be more knowledge intensive than others. Was there

Even is French is the official language, the language competence might be highly diverse. Some people might not understand such a questionnaire well.

Were there incentives? Why would some enterprise contribute?

The discussion is simply a repetition of the results.

The section on implications lists a lot of claims but give very little support for them. I would not agree that this study helps female entrepreneurs or encourages them in any way.

Referencing style is of little help for any reader. Many sentences have a reference attached but do not really tell what the content and contribution of the paper is. Much more detail is needed.

Author Response

(The authors gave the same response as above.)

Round 2

Reviewer 1 Report

Congrats for the revision.

Author Response

Thanks a lot, respected reviewer for your precious time and such a valuable suggestion that purifies and enhances the quality of our manuscript. 

Reviewer 2 Report

The paper was revised substantially

The topic is highly interesting and relevant for society.

Some improvements are visible.

However, many issues remain.

The version which shows the revisions is hard to read, because all format changes are shown on the right

Some of the tables cannot be read at all (page 18 and 19)

The details on recruiting remain vague and need to be strengthened.

Some details on the respondents are now given. I still miss the size of the companies.

Problems with English remain.

Author Response

Respected Reviewer, thank you for your precious time and valuable suggestions. In the revised version author tried to cover English language flaws with appropriate grammar and all the tables were revised with proper settings to make it very clear for the readers. As the author focuses on small-medium enterprises (SMEs) women entrepreneurs who mostly involve their family members and hire fewer employees in their businesses. 

Regards,

Round 3

Reviewer 2 Report

With this file version, the submission could be assessed better.

Still, the problems with English dominate the paper and make it hard to read. It cannot be published as it is.

The research method might be not well selected, I would have opted for a qualitative approach. The concepts are not so clear.

130 references is exaggerated for such a paper.

Not all of my previous comments were addressed.

e.g.: There seem to be over 70 languages spoken in Ivory Coast. I am not familiar with the situation, but just because French is the language of education, it might not be well understood by people with less schooling. It is necessary to write one more sentence about that.

For cultural issues, it could be helpful to check Hofstede.

Author Response

Respected reviewer, the author tried to incorporate all the comments and had proofread from native English twice. We tried our best to meet your reuirements. Thank you so much for your precious time, valuable suggestions, and kind command in purifying our manuscript.

Profound Regards,
